# Are values stable throughout adulthood? Evidence from two German long-term panel studies

Oscar Smallenbroek[1], Adrian Stanciu[2], Regina Arant[3,4], Klaus Boehnke[4]*

**1** Department of Social and Political Sciences (SPS), European University Institute, Florence, Italy, **2** Department of Cognitive Behavioral Sciences, University of Luxembourg, Esch-Belval Esch-sur-Alzette, Luxembourg, **3** Bremen International Graduate School of Social Sciences (BIGSSS), University of Bremen, Bremen, Germany, **4** Bremen International Graduate School of Social Sciences (BIGSSS), Constructor University, Bremen, Germany

* kboehnke@constructor.university

**Data Availability Statement:** The data are held in a public repository [https://osf.io/b7a59/]

**Funding:** Since its inception in 1985 the study referenced below as LuNT Study received occasional minor funding (one-time payments

## Abstract

Previous studies have used cross-sectional or short-term longitudinal data, resulting in a truncated view of a phenomenon unfolding across the lifespan. We find that, contrary to the consensus in the literature, people's values continue developing in adulthood, albeit at a slower pace than in previous developmental stages. We use longitudinal data sources with two measurement instruments. We show their comparability using confirmatory MDS in Study 1 (N = 1,027). We examined value development using latent growth models in a convenience sample of highly educated German peace activists (Study 2, N = 1,209) and corroborated these with evidence from a representative sample from the German population (Study 3, N = 19,566). We find that all values change up to age 40 consistent with theoretical expectations. We observe that with age, self-transcendence and conservation values increase while self-enhancement values decrease. At the same time, we find a curvilinear pattern for openness to change in Study 2 and an overall decrease in Study 3. Moreover, the developmental trajectory of conservation and of self-enhancement in the German general population differ between those with tertiary and without tertiary education. We discuss the implication of the present findings for research on value development and for interventions.

## Introduction

Values are abstract ideals that individuals consider to be important in their lives [1]. Whereas values themselves are inherent to all humans and start to develop early in life, value preferences are specific to an individual and evolve initially in mid childhood, when children emulate the value preferences of relevant others [2, 3]. Specific value preferences crystallize later, namely in adolescence during the phase of identity formation [4]. Although folk wisdom suggests that certain value orientations change across the lifespan (e.g., "People get more conservative with age"), scholars broadly assume that they remain rather stable throughout life. Alwin and

never exceeded 5000 Deutschmark) from Freudenberg-Stiftung, from the German branch of the International Physicians for the Prevention of Nuclear War (IPPNW, Nobel Laureate in 1985), as well as from the Gruner & Jahr and Der Spiegel publishing houses.

**Competing interests:** The authors have declared that no competing interests exist.

Krosnick [5] formulated this as the 'impressionable-years hypothesis;' Inglehart and Welzel [6] refer to the "formative years" in the development of value orientations (in short: value development). However, empirical evidence for value stability from young adulthood onwards is scarce, in our view largely inconclusive, and based on longitudinal datasets that encompass only a short time span [7, 8], but see [9], making it difficult to separate age from cohort effects.

To better understand the relationship between values and age, this paper investigates whether values remain stable across the lifespan or change throughout adulthood with data spanning 18 years. We analyze values of individuals between their (late) twenties and their mid to late forties. Since previous research is highly inconclusive and the literature lacks a developmental conceptual framework [10], we refrain from specifications regarding the direction, magnitude, or facilitators of possible value development. Instead, this paper takes an exploratory approach by analyzing data from two ongoing German panel studies. The LuNT (Life under Nuclear Threat) Study [11] encompasses a sample of highly educated peace movement activists and sympathizers first surveyed in 1985. It measures value preferences since 1999. The GSOEP (German Socio-Economic Panel [12]) surveys the German general population since 1984 and measures values since 1990. Both studies have continuously assessed the value preferences of their participants until 2016/2017. Since the instruments they use are not identical, this paper also presents a methods study examining the degree to which these different instruments measure the same phenomenon, namely value orientations in the theoretical framework suggested by Schwartz [13].

## Value development

Although value theorists propose that values reach lifelong stability in early adulthood [5, 6], empirical evidence would suggest otherwise [9]. This is not surprising because adults experience various transitional events that may modify what they value in life [10], for example entering parenthood [14], experiencing stressful events ([15], Study 4), or migration [16]. Moreover, the aging process itself causes several psychological and physiological transformations that may induce modifications in value preferences [8, 17]. At the same time, with increasing age, it can become more challenging to identify developmental patterns of values because the unique biological and psychological nature of the aging individual is difficult to capture.

These observations correspond with Erik H. Erikson's seminal theory on the stages of psychosocial development, which argues that development is a life-long process with different tasks in successive stages across the lifespan [18]. Erikson suggests that with increasing age the time needed for achieving subsequent developmental stages becomes considerably longer due to the complexity of the motivational conflict characterizing a specific stage. While during childhood developmental phases only last a couple of years, during adulthood they may take up multiple decades, assuming that this may also be the case for value development [19].

**From childhood to young adulthood.** During early childhood the most relevant social context for value development is the family [20–23]. With progressing personality and identity development children gain more independence from their primary socialization context, so that their values increasingly develop in resemblance with their own motivational goals [4, 18]. In accordance with Erikson's theory, studies show that during this time in life, values are least stable, and changes can be observed within short periods of a few months to up to two years [2, 3, 23–25].

However, value development continues throughout adolescence and young adulthood [26, 27]. Studies not only confirm this for different countries (England, Germany, Italy, and Israel), but also for varying time intervals, ranging from a few months up to eight years. Generally,

value development has been found to occur at a low-to-moderate pace with varying stability, although this might depend on the exact life stage. For instance, conservation values seem to be more stable in adolescent years than in young adulthood [26].

**Beyond young adulthood.**   Only a few studies have analyzed longitudinal data in view of value development in adulthood and they mostly cover short time periods, limiting their utility. For example, Bardi et al. ([15], Study 4) examined values of adults in Australia two years apart. They reported power and self-direction as the values with the greatest ($r = .26$) and smallest ($r = .58$) person-specific development. In another study, Bardi et al. ([28], Study 1) measured values of police trainees in England before their training and nine months after. They found evidence in line with a self-selection bias, suggesting that people who hold specific values as important are more likely to develop those values further after a targeted intervention compared to people who hold opposing values as important. Cheung and Lucas [29] used GSOEP data to examine whether values moderated the long-term effects of income on life satisfaction. They collapsed items on having children and having a happy marriage/relationship into an indicator of family values and used an item about career success as indicator for work values. Their analyses for the 1990, 1992, and 1995 data showed that family values peaked around the age of 30 and became less important around the age of 60. However, the results for the 2004 and 2008 waves showed that—after peaking around the age of 30—family values remained rather stable. In contrast, the importance of work values decreases across the lifespan.

Milfont et al. [8] examined value change within a three-year time span with respondents aged 25 to 75 in New Zealand. They extrapolated developmental value curves throughout adulthood and showed that openness to change, self-transcendence, and self-enhancement develop monotonically, whereas conservation values fluctuate around the age of 37 and again around the age of 64. However, the observed data covered only three years, so it still provides a somewhat truncated view on value development across the *individual* lifespan.

Leijen et al. [9] studied changes in values among Dutch individuals aged 16 to 84 over a 12-year period and across four generations. The study found that changes within individuals occurred in all generations, but were more significant among the younger respondents, the Millennials. Millennials increased the importance of benevolence, universalism, self-direction and security while stimulation and power decreased with time. The values of hedonism, achievement and conformity did not change within individuals. This study is unique in its approach, as it is the only one using longitudinal data longer than three years to observe value change in adulthood. However, the authors cautioned against generalizing their findings to other cultural and socio-economic contexts, raising the question of whether the observed value development in adulthood is unique to the Dutch population or has universal significance.

In the absence of longitudinal data, many studies have utilized cross-sectional data to investigate value change across the lifespan. In general, cross-sectional studies on personal values show that with higher age self-transcendence and conservation increase in importance whereas openness to change and self-enhancement decrease [8, 19, 25, 30–32]. However, political science studies looking at the same relationship between values and age interpret these effects as cohort changes. For example, Inglehart and Welzel [6] claim that due to societal economic development conservation values lost relevance for the individual while openness-to-change values increased in their importance. Notwithstanding, without longitudinal data it is impossible to adjudicate between these hypotheses, as the effects due to birth cohort, physical aging, and specific life events cannot be teased apart. In this paper we focus on measuring value change within individuals using the conceptual framework of Schwartz [33] to measure values.

**Value theory and measurement.** Schwartz's [13] Theory of Basic Human Values (TBHV) is one of the most prominent attempts for measuring values today. The TBHV assumes that values represent socially desirable goals. Individuals pursue these goals to address their existential needs and to care for the welfare and survival of their groups [1, 34–37]. Originally, Schwartz proposed ten value types that transcend specific actions and situations: Universalism, benevolence, conformity, tradition, security, power, achievement, hedonism, stimulation, and self-direction (see [38] for 19 more fine-grained value types).

These value types are universal regardless of an individual's personal or social characteristics, such as age, gender, or cultural background. However, people may endorse them to different degrees [13]. Therefore, the crucial issue is the ranking of these values, referred to as *value preferences* or comprehensively as *value orientations*. As displayed in Fig 1, Schwartz organizes value types according to their compatibilities and incompatibilities, highlighting their underlying goals and motivations, also called higher-order values (HOVs) (see also Table 1 below).

The ten value types can be assigned to four different HOVs. The values of security, conformity, and tradition share the goal of creating a safe and predictable environment. They are referred to as conservation values, which are incompatible with openness to change, expressed by hedonism, stimulation, and self-direction, as these values' goals are to explore, create and take risks. Likewise, universalism and benevolence share the goal of transcending the self by caring for and creating relationships with ingroup and non-ingroup members (self-transcendence values), which oppose the goal of gaining a position of strength within society (self-enhancement), as expressed by achievement and power values. It is because of the relationships between goals and motivations as well as possible tensions caused by incompatibilities that values motivate and drive human behavior.

To measure value preferences in accordance with the TBHV Schwartz and colleagues developed two approaches, the Schwartz Value Survey (SVS [13]), and the Portrait Value Questionnaire (PVQ, [39]). In the SVS, respondents indicate to what extent they agree or disagree with 57 different value terms on a scale from -1 (*opposed to my values*) to 7 (*of supreme importance*), Each of these terms addresses a particular aspect of one of the ten value types. This measurement approach is demanding in terms of time and effort and difficult for individuals with a non-abstract thinking style (e.g., children; [1]).

The PVQ offers a solution to these problems [39] as it uses a comparative approach. Schwartz assumes that it is easier for individuals to reflect on their value orientations when they evaluate how similar the motivational goals of fictitious characters are to those, they cherish themselves. Respondents rate character-vignettes regarding their similarity to themselves on a scale from 1 (*very much like me)* to 6 (*not at all like me)*. After reversing scores, value preferences are calculated by aggregating items with the same motivational and goal content of one value type, like for the SVS.

## The present research

The present research is among the first to examine value development in adults over an extended period. We seek to contribute to the literature that builds on the 12-years longitudinal observations by Leijen et al. [9], the developmental curves extrapolated by Milfont et al. [8], and the findings of cross-sectional studies [19, 31].

Following Schwartz's framework of the TBHV, we investigate value development on the level of the parsimonious higher-order structure that arrays the oppositional motivational and goal content of conservation vs. openness to change and self-transcendence vs. self-enhancement. These four higher-order values succinctly capture the main value tensions that are likely to occur across the lifespan. This strategy has several advantages: First, it links with sociological

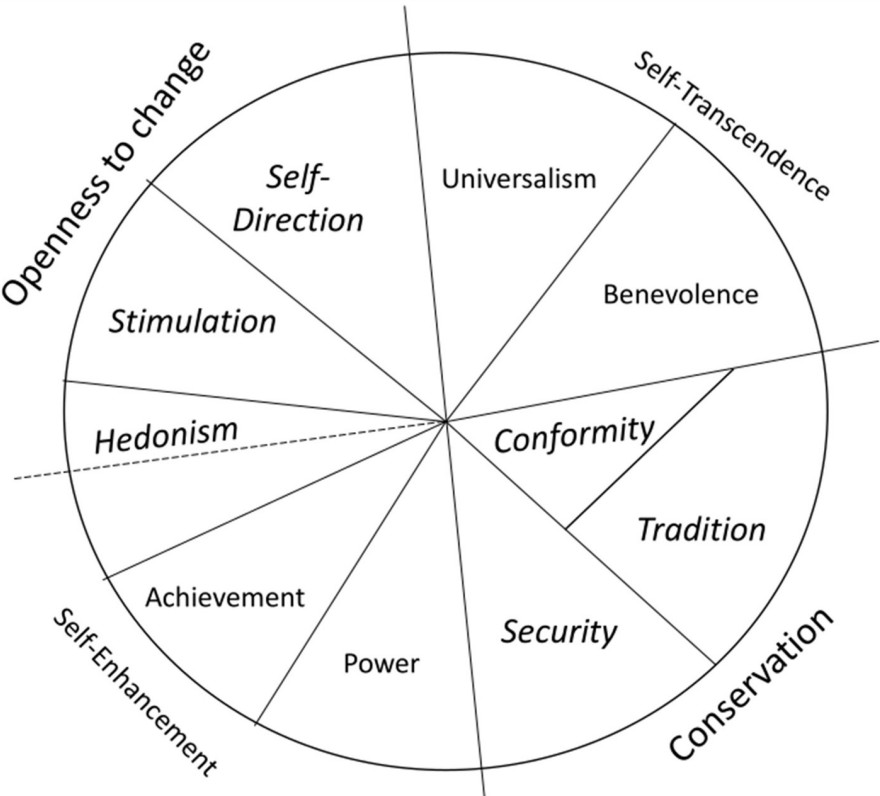

**Fig 1. Schwartz value circumplex.**

research using cohort and social class perspectives on value change [40–42]. Second, it facilitates comparisons with findings on value change from competing value theories, such as the Inglehart-Welzel approach [6, 43, 44]. Third, it avoids resorting to single item measurement which is prone to enhancing measurement error.

Data stem from two longitudinal studies: The LuNT (Study 2) addresses value development in a highly educated snowball-sample of German peace movement activists and sympathizers over a period of 18 years from 1999 until 2017. At the beginning, participants were on average 28 years old (born between 1965 and 1977). Value orientations were measured with a 10-item version of the SVS (abbreviated as SVS-10 here), not to be confused with the Short-SVS, likewise a 10-item instrument [45].

**Table 1. Items measuring Schwartz's higher-order value orientations as depicted in Fig 1.**

| Higher-order value | Kluckhohn-Strodtbeck | SVS-10 |
|---|---|---|
| Self-transcendence | to be socially and politically active; to help others | helpfulness; protection of the environment |
| Openness to change | travel and see the world; fulfil one's potential | pleasure; daringness; creativity |
| Self-enhancement | able to afford something; success in the job | success; social power |
| Conservation | happy partnership; have children | social order; politeness; respect for tradition |

The GSOEP (Study 3), Germany's longest-running panel study with a representative sample of the general population covers an even longer period of 26 years of value assessment. In the first wave, in 1990, participants were on average 21 years old (born between 1966 and 1976). The value orientations of materialism, family life, and altruism were assessed with a modified version of the Kluckhohn-Strodtbeck instrument [46]. We include the GSOEP to contrast the LuNT Study with representative data. However, using the GSOEP data for the present paper comes with one central drawback: It uses the Kluckhohn-Strodtbeck instrument (abbreviated as K-S here), which is not directly related to the TBHV. However, Schwartz was strongly influenced by earlier work in the field: Kluckhohn [47] proposed that the number of existing value types is limited. Florence Kluckhohn and Strodtbeck [48] developed this thought further and defined values as *concepts of the desirable* in relation to how societies respond to existential problems.

In Study 1we first examine the degree to which they measure the same phenomenon, namely value orientations in accordance with the theoretical framework suggested by Schwartz [13].

## Transparency and openness of our research

We collected data in an online panel for Study 1 to create a covariance matrix of all items used in Study 2 and Study 3, which both rely on secondary data sources. The sample size of Study 1 was determined in ways required to fulfill minimal sample size for statistical inference (in our case 16 German federal states by two genders, the two quota variables, leading to a necessary $N$ of 960 to 1280). Across the studies there was no deception involved. All measurement instruments are reported in detail, and all study materials including collected data, syntax, and supplementary material (SM) are available at https://osf.io/vjpr4. Data for Study 1 were analyzed using R, Version 4.1 and the package *smacof*, whereas data for Study 2 and 3 were analyzed using Stata 15.

## Study 1: Convergent validity

We assess the convergent validity of the K-S and SVS-10 to test whether these instruments measure the designated theoretical construct: (higher-order) value preferences as theorized by Schwartz. Construct validation theory [49, 50] states that any psychological construct that is not immediately tangible, like values, manifests itself across observable situations which are quantifiable with an appropriate instrument. One strategy to assess the validity of a given instrument is to establish convergent validity and compare it with an established instrument for which construct validity has already been shown, in our case the PVQ-21.

The PVQ-21 is used in the European Social Survey (ESS) and has been shown to have good construct validity [51]. It therefore serves as the benchmark. The K-S and SVS-10 differ from instruments developed by Schwartz in several ways. In contrast to the PVQ-21 they ask respondents directly whether short phrases or concept terms are important to them. Besides the differences in question wording, each instrument has different response options and uses slightly different content. However, at face value both are covering all four HOVs with at least two items. We assess the similarity between the three questionnaires using exploratory and confirmatory multidimensional scaling (MDS) [52]. MDS is customary in the values literature to assess the structural relationships between items [13, 38].

## Method

### Participants

Data from 1,030 participants were collected in mid-2019 in an online quota survey from all 16 German federal states by *respondi*, a private company specialized in market research. Characteristics of the sampled population were pre-defined to match those of the LuNT and GSOEP studies as closely as possible. Participants' age ranged from 30 to 50 with a median age of 40. One half of the participants were women ($N$ = 515), the other half were men. All participants responded to the PVQ-21, SVS-10, and K-S in German. The PVQ-21 and K-S items were taken directly from the German versions of the ESS and the GSOEP, respectively. The SVS-10 items were identical to those in the LuNT Study.

For quality control, cases with missing information on more than 75% of the items of an instrument or cases using the same response to at least 75% of items within an instrument, across all three instruments were dropped from analyses (also see [51]). In total, three cases were dropped due to response pattern, reducing the final sample size to $N$ = 1,027.

### Measures

**PVQ-21.** Participants read 21 short descriptions about fictitious male or female characters (depending on their own gender) who were introduced in terms of what they hold as important in life. Each description referred to one value type in the TBHV. There were three items for universalism and two for the nine other value types. Participants had to indicate how similar the fictitious persons were to them (1-*very much like me*, 6-*not at all like me*). An item example was: "Being very successful is important to her[him]. She[He] hopes people will recognize her[his] achievements" (see S1 File).

**SVS-10.** Single items with a 9-point Likert response scale (-1 *contrary to my values*, 0 *not important* to 7 *very important*) assessed preferences for each of the ten basic values in the TBHV. Participants had to rate the importance of each of the following values in the order they were present in the LuNT Study: willingness to help (benevolence), social order (security), success (achievement), protection of the environment (universalism), politeness (conformity), creativity (self-direction), pleasure (hedonism), respect for tradition (tradition), social power (power), and daringness (stimulation). The item intro read: "Below you find 10 values that can be important to people. Please mark for each of these values how important it is for you."

**K-S.** The K-S was intended to measure three life goals: success (materialism), family life, and altruism. Participants were asked "Different things are important to different people, how important are the following things to you?" and to rate them on a scale from 1-*very important* to 4-*not at all important* (cf. [46]. The K-S includes the items: affording something, fulfilment, career success, travel and seeing the world, owning a house, happy marriage/relationship, having children, being there for others, and social/political involvement. Also here, scoring was reversed for the subsequent analyses.

**Demographic variables.** This study asked respondent's gender, age and educational attainment. Education was recorded using seven response options (1 = less than primary school) up to graduate level (7 = PhD).

### Analytical strategy and procedure

MDS attempts to identify structure in a set of (dis)similarities between objects and visualise them for analysis. In our case, we used a correlation matrix of the item ratings (MDS analysis on ipsatized items are documented in S2 to S5 in S1 File). According to the TBHV, four areas should emerge, structured around two axes. The first axis should separate social from personal

goal-focused values, the second axis should separate growth from anxiety motivated values. Combining these two axes should create the four HOV orientations.

To assess whether the SVS-10 and the K-S assess human values similarly to PVQ-21, we randomly split the data into two subsamples. Using the first subsample we applied an MDS without constraints. However, MDS algorithms often select non-optimal solutions. Therefore, we ran 1000 MDS using random starting configurations and plotted the best fitting solution as proposed by Borg et al. [52].

The second subsample was subjected to a confirmatory MDS [52] which allows the user to define a theoretically derived starting point. Items from the K-S and SVS-10 were chosen to form HOV based on their location in the exploratory MDS. These measures were added to a correlation matrix of the PVQ-21 items. It was then possible to assign positions for each variable following TBHV (S1 Table A and Table B in S1 File). The resulting MDS elucidated whether the HOVs correspond to the 'correct' value items in the PVQ-21 shown in Fig 1 (see also S5 Fig J in S1 File for ipsatized results).

For each MDS, we evaluated the stress using the Stress-1 function and performed a permutation test. The Stress-1 function is an algorithm which shows the dissimilarity (stress) between the correlations in the data and the distances in the MDS solution. The higher the stress, the lower the fit, thus solutions with lower stress are preferred. The permutation test shows whether the stress is significantly different from random permutations of the data, mimicking traditional significance testing. We assessed the stability of solutions with the jack-knife (outlier sensitivity check) and bootstrap procedures (for 95% confidence intervals around the coordinates) which are available in the S1 File. All MDS analyses were conducted in *R* v.4.1 using the *smacof* package [53], specifying ordinal scaling and two dimensions.

## Results

### Exploratory MDS of K-S, SVS-10, and PVQ-21

The exploratory MDS solution showed a good separation of items into HOVs. Stress-1 was 0.22 and significantly lower ($p < 0.001$) than solutions based on random permutations of the data. The PVQ-21 items create five areas corresponding to three HOVs. The openness to change items belonging to self-direction were separated from stimulation and hedonism items. The SVS-10 items are all found close to their respective PVQ-21 items. The only exceptions are the conformity and security items which are between PVQ-21 items belonging to self-transcendence and conservation. The K-S items are also located near their respective PVQ-21 items. However, the three items on conservation (partner, children, house) are between PVQ-21 items on conservation and self-enhancement. The item "owning a house" is closer to self-enhancement items than conservation items. It could equally be interpreted as a security or conformity value, as well as a sign of social status, power, and achievement and therefore we exclude it from further analysis as a precaution.

### Confirmatory MDS of K-S, SVS-10 and PVQ-21

Since both the K-S and the SVS-10 showed a satisfactory separation of the items into areas interpretable as HOVs like the PVQ-21, we were able to assess the similarity between the three value measures by performing another confirmatory MDS. Entering all (single) PVQ-21 items as well as the HOV scores for the K-S and SVS-10 revealed a solution that was a good and stable approximation of the correlation matrix (S5 Fig J in S1 File). Stress-1 was 0.13 and significantly lower ($p < 0.001$) than solutions based on random permutations of the data.

Fig 2 plots the MDS solution. It indicates that K-S and SVS-10 measured self-transcendence and self-enhancement values well. The HOVs are grouped in a cloud among the respective

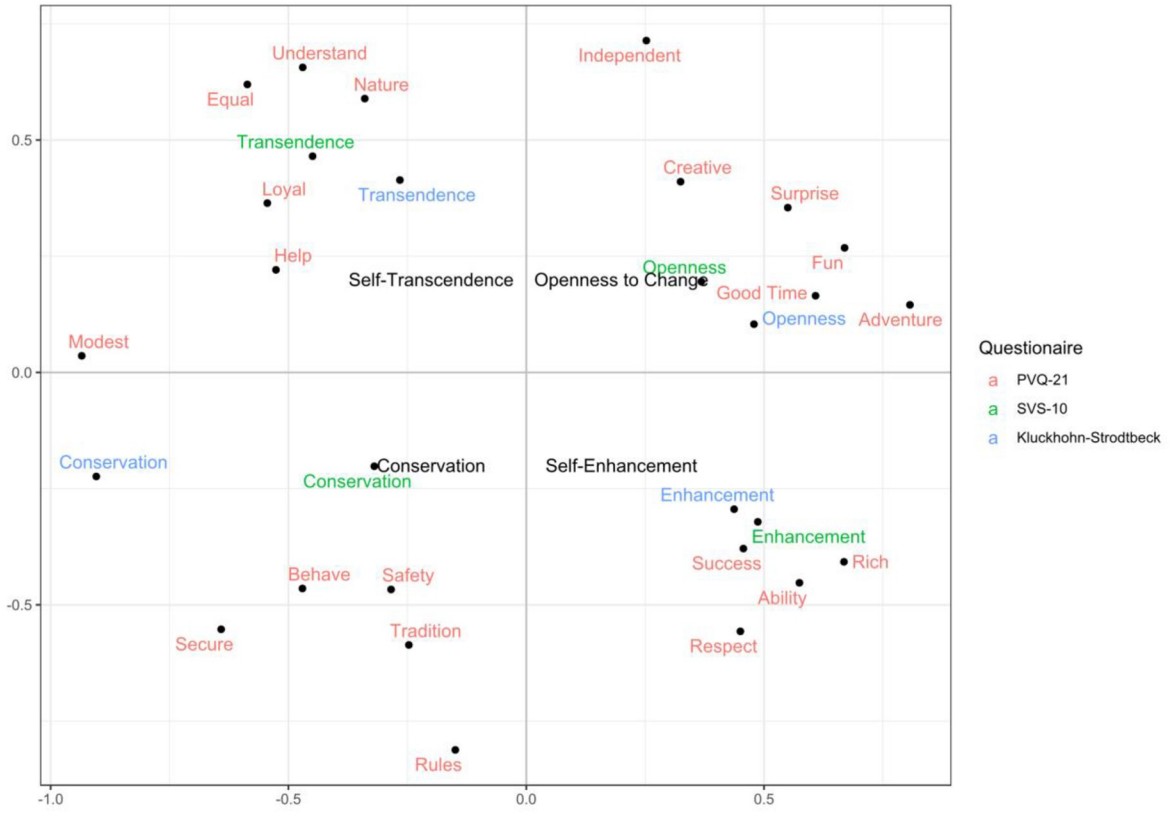

**Fig 2. Confirmatory MDS of PVQ-21 items and HOVs from the K-S and SVS-10 instruments.** *Note.* MDS projection of PVQ-21 item ratings, Kluckhohn-Strodtbeck and SVS-10 higher order value ratings. The MDS projection uses a theoretical starting configuration. Data from Study 1, Split Sample 2.

PVQ-21 items. Second, openness to change of the K-S and SVS-10 are located closer to the stimulation and hedonism items of the PVQ-21 than to the self-direction items (independent and creative). Nonetheless, both HOV measures are in the appropriate quadrant of the value circle. This indicates that the openness to change measurements in this study may not generalize all that well to studies emphasizing self-direction values as primary openness values. Third, conservation was measured well by the SVS-10 since it is located in proximity to all the relevant PVQ-21 items. In contrast, the conservation measure of the K-S was located closer to the 'modest' item than to the other conservation items of the PVQ-21. This shows that the K-S and the SVS-10 measurements of conservation emphasize different aspects of the HOV orientation, however, both are clearly good measurements of conservation as their position is closest to the PVQ-21 conservation items and in the appropriate quadrant of the graph.

## Discussion

To the knowledge of the authors this study was the first to investigate the convergent validity of the K-S and the SVS-10 against the PVQ-21. The results of our confirmatory MDS analyses show that the convergent validity of both instruments is acceptable. This means, they measure what they are supposed to measure, namely the four HOVs postulated by the TBHV. A reassuring finding given that the SVS-10 was developed by the fourth author in close alignment with the TBHV. Our findings provide initial evidence on the convergent validity between the

K-S value instrument as used in the GSOEP and the PVQ-21. Value researchers can now apply Schwartz's HOVs typology to the GSOEP dataset and examine psychological mechanisms that link human values to thought, emotion, and behavior in a lifespan perspective. Moreover, our approach for testing construct validity can inform users of other large survey programs on how to re-purpose existent data for value research.

Nonetheless, some limitations are worth discussing here. Our analysis detected a deviation from Schwartz's value circumplex model that is quite common for the PVQ-21 and SVS-10: Self-direction items are separated from stimulation and hedonism items. This suggests that the measurement of openness to change values as assessed with the SVS-10 in the LuNT Study is more likely to reflect hedonism and stimulation values than self-direction values. Yet more problematic is the fact that the item probing the life goal of owning a house in the GSOEP data assessed by the K-S seems to measure a mix of conservation and self-enhancement values. Since such a double-barreled item would impede the interpretation of the value development curves from the GSOEP data, we decided to drop it from any further analyses. We are confident that through this step we can study value development throughout adulthood by reproducing Schwartz's value circle with the K-S and the SVS-10, as suggested by the TBHV.

## Study 2: LuNT–German peace activists

Study 2 sets out to examine how value preferences develop across the lifespan in a sample of highly educated German peace activists and sympathizers.

A team of researchers in West Berlin started the LuNT Study in 1985 with support from the International Physicians for the Prevention of Nuclear War (IPPNW), who were awarded the Nobel Peace Prize during that year ([11]). Barred by Berlin state authorities from conducting school-based representative sampling, the researchers were forced into gate-keeper-guided snowball sampling. Supported by calls in local media outlets and private networks, they asked members of the community (teachers, NGO activists, etc.) to assist in distributing a four-page questionnaire to children and adolescents in West Germany. This strategy resulted in 3,499 completed questionnaires. In total, 1,492 participants left their addresses to be followed up at a later point. Data collection occurred henceforth every 3 ½ years. The study began assessing values of participating individuals in its fifth wave in 1999, when their average age was 28. To the knowledge of the authors, the LuNT Study is the largest ongoing longitudinal survey containing data on human values measured in line with the TBHV.

## Method

### Participants

The core sample provided data in each wave from 1999 (Wave 5) to 2016/17 (Wave 10). Wave 11 data from 2020 have been gathered but could not yet be included here. At Wave 5, participants were on average 28 years old ($SD$ = 2.73), enrolled in tertiary education (38.80%), either part-time or full-time employed (68.50%), single (65.30%), and had no children (81.00%). At Wave 10, participants were on average 46 years old ($SD$ = 2.79), with at least tertiary education finalized (60.70%–13% had a doctoral degree), employed (78.10%), married (53.30%), and had children (62.80%). We dropped two cases in Wave 9 (2013), where respondents indicated all values to be "not important". The remaining 242 respondents (142 women, 100 men) were included in the main analyses, at least 184 respondents contributed information in 1999, in 2017 and at least two more occasions.

## Measures

Values were measured using the ratings on SVS-10 instrument to form HOVs (see Study 1). Items were re-coded so that they range from 0 to 8 instead of -1 to 7.

The *year* variable ranges from 1999 to 2017. However, the convergence of maximum likelihood estimation is aided if variables have a mean of 0 and a small standard deviation. Therefore, we rescaled the *year* variable, so that 1999 = 0 and a one-unit increase is set to 4 calendar years. The integer closest to the actual measurement interval was chosen for rescaling in order to facilitate smooth maximum likelihood estimation. As survey waves were interspersed by 3.5 years, the time points in the data are 0, 0.875, 1.75 etc.

## Analytic strategy and procedure

We first report descriptive statistics of the mean level endorsement of each HOV in 1999 (Time 1), their rank order stability between waves and their intraclass correlations (ICC). The mean level of each HOV can be compared to examine the average value preferences in the sample. The rank order stability is the correlation of HOV between two time points. A high rank-order stability indicates that, on average, respondents stay within the same percentile of the population in terms of the HOV rating between time points. ICCs divide variation in HOV into a between and within-person component. A high ICC indicates that the responses a person gave across the years of participation are like each other, whereas a low ICC indicates that HOV scores did differ substantially when looking at the scores obtained from one and the same person across the years.

We employed multilevel modeling in Stata 15 using the *mixed* command to estimate the mean intercept (the general overall level) and slopes (change gradients independent of the overall level of scores) of each HOV from Wave 5 to Wave 10. Multilevel models can be used to create growth models and are appropriate when data have a nested structure with repeated observations (time points) within higher units of analysis (in our case individuals). We began with a model including only a random intercept to estimate the intraclass correlation. We fitted additional models to account for value change with a linear, a squared, and a cubic fixed slope. In case the fixed slope was significant, we assessed whether there were individual differences in the slope by adding a random slope.

We assessed the model fit using three statistics, namely the AIC, BIC, and the Likelihood Ratio test (S7 Table G in S1 File). Like the AIC, the BIC assesses overall model fit, but emphasizes parsimony by introducing a penalty for each parameter [54]. Likelihood Ratio tests provide a statistical test ($\chi^2$) of the difference between the log-likelihood of two models. In the results section, we will comment on the best fitting model and report the standardized regression coefficients (*ß*) of the year variable and show the predicted means of each HOV in figures.

## Results

### Value priorities, rank order stability, and intraclass correlations

Table 2 summarizes mean levels of HOVs at $T_1$ (1999) in the second column and the rank order (correlations between value measurement occasions) in the third to seventh column. In 1999, when participants were on average about 28 years old, they attributed by far the greatest importance to self-transcendence, followed by openness to change, conservation, and self-enhancement. In the 3 ½ years between each measurement, the mean levels of all HOVs showed moderate to high rank-order stability. While openness to change became more stable with each new measurement, conservation continuously lost stability over time. Interestingly, self-enhancement remained rather stable across the years except between $T_2$ (2002/3) and $T_3$

**Table 2. Means, standard deviations, and longitudinal correlations of hovs in the LuNT data.**

|  | M (SD) $T_1$ | $T_1$-$T_2$ (N) | $T_2$-$T_3$ (N) | $T_3$-$T_4$ (N) | $T_4$-$T_5$ (N) | $T_5$-$T_6$ (N) |
|---|---|---|---|---|---|---|
| Openness to Change | 4.74 (1.42) | .55 (194) | .54 (171) | .64 (179) | .61 (175) | .67 (166) |
| Conservation | 4.64 (1.29) | .69 (193) | .68 (171) | .61 (179) | .63 (175) | .54 (166) |
| Self-Enhancement | 3.14 (1.36) | .63 (193) | .67 (170) | .61 (178) | .58 (175) | .60 (166) |
| Self-Transcendence | 6.24 (1.16) | .58 (194) | .69 (171) | .67 (179) | .51 (175) | .48 (167) |
| Mean Age | 28 | 28<>32 | 32<>35 | 35<>38 | 38<>42 | 42<>46 |
| Observations |  | 193 | 170 | 178 | 175 | 166 |

*Note.* $T_1$ = 1999. Further data collection every 3 ½ years (2002/3, 2006, 2009/10, 2013, 2016/7). All correlations significant at $p < .001$

(2006), when participants were on average about 32 and 35, respectively. Self-transcendence showed the most dynamic pattern: This HOV increased in rank order stability between the late 20s and the mid-30s (from $T_1$ to $T_2$ to $T_3$), and then gradually decreased during the subsequent measurements when respondents entered their 40s.

Intraclass correlations (ICC) of the four HOVs were quite similar, ranging from 0.55 to 0.58, all with a standard error of .003, indicating a moderate correlation between observations within each person (see Table 3). This means that over half of the variation in the sample can be attributed to differences between persons, whereas a little over 40% of the variance is attributable to within-person change. A decomposition of the explained variance shows that the random slopes (time trends) explain a small portion of the variance, 2.8% of openness to change and up to 5.5% of self-transcendence.

## Value development trajectories

Self-transcendence is best modeled with a random intercept and a random linear slope. The AIC and BIC both indicate that adding a squared slope would improve model fit. However, we decided against adding a squared slope because the estimated standard error of this coefficient indicates that its estimation is not reliable (in comparison to the coefficient their S.E. are very large). The L.R. test supports our decision to keep a simpler model through a non-significant increase in the log likelihood after adding a squared slope to the model. The large S.E. and the non-significant increase in the log likelihood indicate that a squared slope may be overfitting the data. The chosen model indicates that individuals vary significantly regarding the level of endorsement in self-transcendence and in the change gradient (slope). The chosen model shows that, on average, the importance of this HOV increases steadily across years ($\beta = 0.063$, $p = 0.009$). Fig 3 illustrates the steady increase in self-transcendence from the respondents' late 20s (estimated mean = 6.19) to their mid to late 40s (estimated mean = 6.41).

We found openness to change to best be modeled with a random intercept, a random linear slope, and a fixed squared slope. This indicates that there is substantive variation in the level (intercept) and the linear component of change over time (slope) between individuals, however there are too few differences to warrant individual differences (random effects) in the

**Table 3. Inter class correlations and S.E. of higher order values in the LuNT data.**

| Higher Order Value | Inter Class Correlation | S.E. | Lower Bound | Upper Bound |
|---|---|---|---|---|
| Self-transcendence | 0.55 | 0.03 | 0.49 | 0.61 |
| Openness to change | 0.57 | 0.03 | 0.51 | 0.63 |
| Self-enhancement | 0.57 | 0.03 | 0.51 | 0.62 |
| Conservation | 0.58 | 0.03 | 0.52 | 0.63 |

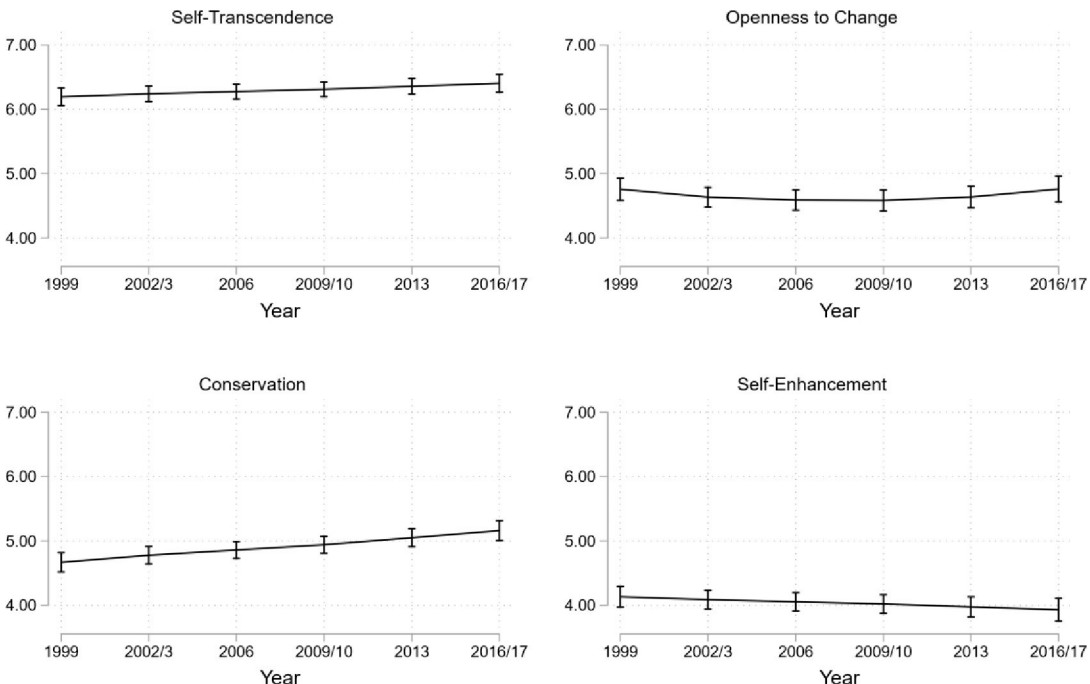

Fig 3. Estimated means of higher-order values in the LuNT study.

squared slope. We find a negative linear slope ($\beta$ = -1.65, $p$ = 0.009) and a positive squared slope ($\beta$ = 1.71, $p$ = 0.006), indicating a parabolic shape to the overall value development as displayed in Fig 3. In 1999, when respondents were on average about 28 years old, the estimated mean rating of openness to change equals 4.76, drops to 4.58 ten years later, and increases again to 4.76 in 2017 at the last data point when respondents were in their mid-forties.

Self-enhancement is also modeled best with a random intercept and random linear slope, indicating that there is substantive variation in the level and change over time between individuals. We find a negative linear slope ($\beta$ = -0.05, $p$ = 0.028), indicating that self-enhancement steadily loses importance with increasing age. The estimated rating of 4.13 in 1999 (Wave 5) decreases to 3.93 in 2017 (Wave 10). Fig 3 displays the trajectory.

Likewise, conservation is best modeled with a random intercept and random linear slope. The random effects in the model indicate substantive variation in the level of conservation between individuals and differences in the amount of change over time between individuals. The average slope is linear and positive ($\beta$ = 0.13, $p$ < 0.001), indicating a constant increase towards a higher appreciation of conservation with each time point. This means that conservation continuously gains importance for adults between their late 20s and the mid to late 40s (see Fig 3), moving from an average endorsement of 4.67 to 5.16.

## Discussion

Overall, we found all HOVs to exhibit high stability in rank order and moderate stability in the mean ratings between 1999 to 2017, like the longitudinal studies based on 2–8-year time spans (for a review see [7]) and over a 12-years period [9]). Does that mean that values indeed remain stable in adulthood, as suggested by Ron Inglehart [55]? The answer, in our view, is no.

In fact, we find a significant linear and/or squared trend for each of the four HOVs, suggesting that, although no extraordinary modifications could be observed, noticeable

changes over time occurred for all of them. The ICCs corroborate this interpretation, showing that around 45% of the variation in HOVs is attributable to within-person change over the studied 18-year period, however the random slopes explained only around 5% of the variance. This result underscores Erikson's notion that after adolescence development slows down but never stops [18].

Our findings for openness to change illustrate the continuous change and highlight differences found over a period of 18 years compared to shorter time spans. In their study from New Zealand, Milfont et al. [8] found a downward trajectory for this HOV across ages 25–73, based on three-year-interval data. Transferring this result to the present study with a much longer time span would have meant expecting a constant devaluation of openness to change between the late 20s and late-40s. However, at least among the sample of highly educated peace activists and sympathizers from Germany, this trajectory was not found. Our analysis rather suggests a parabolic trend, meaning that the importance of openness to change decreased after the initial measurement, remained weaker during the participants' 30s and returned to the initial levels in the early 40s. Furthermore, our analysis shows a greater magnitude of change, between -0.20 and 0.20, compared to Milfont et al. [8] who found between –0.04 and –0.10 in respondents of similar ages and on a comparable 8-point scale.

Our study also provides evidence that results on value development from cross-sectional surveys and shorter longitudinal studies confound cohort and age-related effects [8, 31, 56, 57]. For example, mean level changes in openness to change and self-transcendence estimated with the LuNT Study substantially differ from those reported by Milfont et al. [8], who find an overall decrease in openness to change and a much steeper increase in self-transcendence.

Of course, we can only speculate what was responsible for this development in the lives of these study participants during their 30s, but focusing on career and starting a family might very well lead to a (temporary) decrease in the importance of openness to change, simply because individuals lack the capacity for it during this demanding time of their lives. Furthermore, we found a substantial heterogeneity between individual participants in both the mean level and change gradient. The standard deviation from the mean slope was around 0.17 for the four HOV values. The variation in slope would be interesting to pursue in future studies using socio-economic, demographic, or lifespan research approaches.

Compared to Milfont et al.'s study [8] another finding of the present research is remarkable, namely that the slope for conservation values is positive and not negative. This divergence might occur due to differences in the studied populations. An increasing preference for conservation values seems rather counter-intuitive for a sample of highly educated peace movement activists and sympathizers. Yet, the participants were active in the protests that took place in the context of a looming nuclear threat during the Cold War. Germany at the time was a divided country between its western part under the influence of the U.S. and its eastern part that was controlled by the U.S.S.R. Conservation as in preserving the social order, being polite, and showing respect for tradition might carry a different meaning for these study participants compared to the general population of New Zealand where the fear of a nuclear apocalypse was unlikely in 2009 compared to Germany at the time.

Its sample characteristic is a general weakness of the LuNT Study. Data were collected via snowball sampling from individuals who were interested in the peace movement and actively decided to participate in the study over decades. Therefore, drawing inferences from the data to the broader population is impossible. This is the reason for turning to Study 3 next, which examines value development over a similar period of time in Germany using a representative sample.

## Study 3: German general population

Following Study 2, we model intra-individual change over a large timespan. Given that Study 2 participants were almost exclusively highly educated and thereby an unrepresentative sample, we will also examine whether education level plays a role in value development in adults. Studies found positive associations with self-transcendence [58] and openness to change values, and a negative relationship with conservation ([13, 38]. However, we expect that our analyses with general population data from the GSOEP will otherwise replicate the findings from the LuNT study, namely that self-transcendence and conservation values increase their importance across the lifespan, conservation more strongly so than self-transcendence. At the same time, self-enhancement should decrease in its importance, while openness values should follow a U-shaped change trend.

## Method

### Participants

The GSOEP is a representative longitudinal household survey of the German general population. Since its launch in 1984 interviews are conducted annually and face-to-face with all members of a household aged 16 and older. However, the K-S was only fielded in eight waves, namely in 1990, 1992, 1995, 2004, 2008, 2010, 2012, and 2016. To compare the GSOEP and the LuNT data, a sub-sample matching the demographics of the LuNT study was drawn. We selected participants who responded to the K-S, belonged to similar birth cohorts as the participants of the LuNT Study, and lived in West Germany in 1989.In the LuNT Study participants were born between 1965 and 1977. In the GSOEP subsample we used, participants were born between 1966 and 1978.

The GSOEP periodically draws refreshment samples [59] to decrease statistical biases due to dropout and non-response. In contrast to the LuNT data, the GSOEP contains respondents that were not observed over the entire study but still provide data spanning 26 years. The two main samples are the West German sample which includes respondents who answered in each wave from 1990 to 2016, except 2010, and the Cohort/Family Types sample which started in 2010. However, most respondents in the sample provide data either from 1990 to 2008 or from 2010 onwards (N = 6103, 78%) whereas 22% of (N = 1730) respondents provide data spanning the entire period from 1990 to 2016. For more information see S9 Fig L in S1 File.

The GSOEP encompasses 20,474 eligible observations from eight waves between 1990 and 2016. Of those, 908 observations were dropped because of missing data on the K-S value measure. Thus, the analytical sample contains $N$ = 19,566 observations from 7,833 individuals. On average, respondents participated two to three times; 37% of the sample were surveyed more often. More information on the sample size in each wave in S9 Table J and Fig L in S1 File. Age ranged from 14 to 24 in 1990 ($M$ = 20.92, $SD$ = 2.26) and from 38 to 50 in 2016 ($M$ = 44.66, $SD$ = 3.66). It is important to emphasize that the samples analysed in Study 2 and Study 3 match concerning their birth years (cohort), but *not* regarding their age at the first time of measurement. While LuNT participants were first surveyed in 1999 (Wave 5) around the age of 28, GSOEP participants were surveyed earlier, namely in 1990 ($T_1$), at about 21 years old. However, since data of the last analyzed data point were collected in 2016/17 in both studies, the average age at the end of the panels is similar for both samples.

## Measures

Value ratings were aggregated to the HOV measures from the K-S as described in Study 1. Education was coded 1 if the respondent obtained general or vocational tertiary education (33%) at any time in the panel and 0 if they obtained no tertiary education (67%).

The *year* variable ranges from 1990 to 2016. To match the procedure in the LuNT Study, it was rescaled, so that 1990 corresponds to 0 in the data, with one unit standing for four calendar years.

## Analytical procedure

This study assesses value trajectories in the German general population using the GSOEP data following the steps described in Study 2. In contrast to Study 2, we included a variable for higher education. First, we will present the average ratings of each HOV in 1990, the rank order correlations between waves, and the ICC. Then we report on the best fitting models and on the estimated level of value ratings (intercept) and change in value ratings over time (slope) of each HOV which are illustrated in the figures below.

The strategy for model selection increases the model complexity stepwise. Beginning with the assessment of the shape of the time slope, we first modeled each value using a random intercept to estimate the intraclass correlations and then added fixed and random effects of time. Next, we added interactions of time to model non-linear changes in the HOV, then added a fixed effect of education and in subsequent models an interaction between education and linear and non-linear effects of time (S10 Table K in S1 File). The fit of each model was assessed with AIC, BIC, and LR (S10 Table L in S1 File).

## Results

### Value priorities, rank order stability, and intraclass correlations

Table 4 documents mean levels of HOVs at $T_1$ as well as the rank-order correlations between measurement occasions. In 1990, when participants were on average 21 years old, they attributed the greatest importance to self-transcendence, followed by self-enhancement, and openness to change. Conservation ranked lowest. Stability between measurement occasions regarding the mean-level importance of HOVs was overall moderate (about $r = .50$, $p < .001$). Openness to change showed a slight dip in stability between 1990 and 2004, to increase afterward until 2016. Conservation as well as self-enhancement values, showed an increase in stability over time until the end of the observation period when study participants were in their mid to late 40s. Self-transcendence being the most important value for the participants, on average shows the lowest rank-order stability initially but gains stability over time.

Intraclass correlations (ICC) of the four HOVs in the GSOEP data were rather similar to each other. The pairs of HOVs on opposite sides of the Schwartz value circumplex exhibit similar and moderate ICCs: self-transcendence and self-enhancement both .45, openness to change and conservation .48 and .50, and each of them with a standard error of .01. This suggests that for self-transcendence and self-enhancement some 45% of the variation are attributable to differences between individuals, whereas 55% stem from variation within individuals. In contrast, openness to change and conservation are roughly evenly split in terms of between and within individual variation. The decomposition of the explained variance shows that the random slopes explain about 5% of the variation in openness to change, self-transcendence and self-enhancement values but substantially more variation of conservation (10%).

Table 4. Means, standard deviations, and longitudinal correlations of HOVs in the GSOEP data.

| | 1990 Mean (SD) | | Correlations between Waves | | | | | |
|---|---|---|---|---|---|---|---|---|
| | $T_1$ ($N = 1{,}011$) | $T_1$-$T_2$ $N = 878$ | $T_2$-$T_3$ ($N = 1{,}002$) | $T_3$-$T_4$ ($N = 912$) | $T_4$-$T_5$ ($N = 1{,}983$) | $T_5$-$T_{6/7}$ ($N = 1{,}897$) | $T_7$-$T_8$ ($N = 3{,}289$) | |
| Openness to Change | 1.97 (.59) | .55 | .52 | .47 | .56 | .55 | .65 | |
| Conservation | 1.77 (.64) | .46 | .59 | .50 | .68 | .67 | .72 | |
| Self-Enhancement | 2.02 (.48) | .43 | .49 | .49 | .58 | .67 | .64 | |
| Self-Transcendence | 2.41 (.49) | .35 | .56 | .32 | .49 | .49 | .56 | |
| Mean age | 21 | 21<>23 | 23<>26 | 26<>35 | 35<>39 | 40<>42 | 42<>46 | |

*Note*. Value preferences have been recorded in 1990, 1992, 1995, 2004, 2008, 2010, 2012, 2016. In 2010 respondents from the Families in Germany Survey was integrated into the GSOEP and answered the Kluckhohn-Strodbeck questionnaire. However, the regular GSOEP respondents did not. Therefore, there are no respondents with value measures observed in both 2008 and 2010. All correlations are significant at $p < .001$. The N increases across waves due to refreshment sampling (see S9 Fig L in S1 File).

## Value development trajectories

We ran several multilevel models to assess which trajectories best describe the change in HOVs in the GSOEP data. Based on the three model fit criteria identical models were selected for conservation and self-enhancement. They include a random intercept and random slope. This indicates that there is substantial variation in the extent to which individuals endorse these HOVs (intercept) as well as variation in the change trends over time (slope). Additionally, we found that a fixed effect of year-squared and of year-cubed improved model fit, indicating that there is non-linear change in conservation and self-enhancement values, but that there are too few individual differences in the magnitude of non-linear change to warrant random effects. Finally, we included a fixed effect of tertiary educational attainment to model differences in the intercept of these HOVs. Models with an interaction between education and year had a worse fit, indicating there are no differences in the slope of these HOVs between education levels (S10 Table L in S1 File). For openness to change and self-transcendence, the best fitting model is the same as described above, except that it includes an interaction term between education level and the year variables. This indicates that the non-linear slopes of these HOVs differ significantly between education levels, meaning that different education levels exhibit different change trends.

The overall trajectory of self-transcendence values is shown in Fig 4. Preferences of this HOV increase over time from the first wave in 1990 to the final wave in 2016. The estimated means of self-transcendence increased from their initial level of 2.60 for participants without tertiary education and 2.75 for the participants with tertiary education to reach 2.75 and 2.90, respectively, in 2016. However, the mean level changes are not constant. Little change in the estimated average between 1990 and 1995 is followed by lager increases until 2004 and further smaller increases until 2016. Fig 5 shows that the change gradient (slope) is not constant. The slope is negative in 1990 ($\beta = -.27$, $p < 0.001$, see S11 Fig N in S1 File), becomes indifferent from 0 by 1995, and reaches a peak in 2004 ($\beta = .22$, $p < 0.001$) after which it decreases again to non-significance by 2016.

The overall trajectory of openness to change is illustrated in Fig 4 (S12 Table N Table O Table P in S1 File for estimated coefficients). It shows a decrease in the importance of this HOV which seems to level out around 2010. Additionally, better educated respondents rate this value to be more important than lower educated participants (1990: 3.16 vs. 3.00; $p < 0.001$; 2016: 2.73 vs. 2.57; $p < 0.001$). Fig 5 shows that the openness to change slope is non-linear, it starts off strongly negative in 1995 ($\beta = -0.31$, $p < 0.001$), but increases slowly to become positive by 2016 ($\beta = 0.09$, $p < 0.015$).

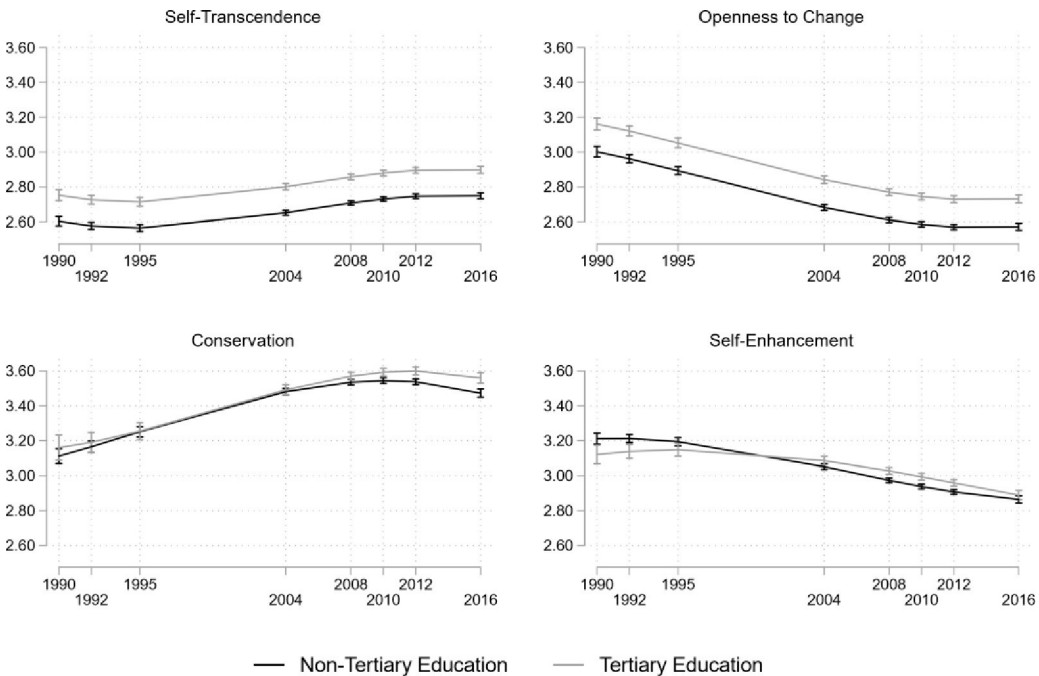

**Fig 4. Estimated means of higher-order values on a 4-point scale in the GSOEP data.** *Note*. Bars indicate 95% confidence intervals. Black lines stand for people without, grey lines for people with tertiary education.

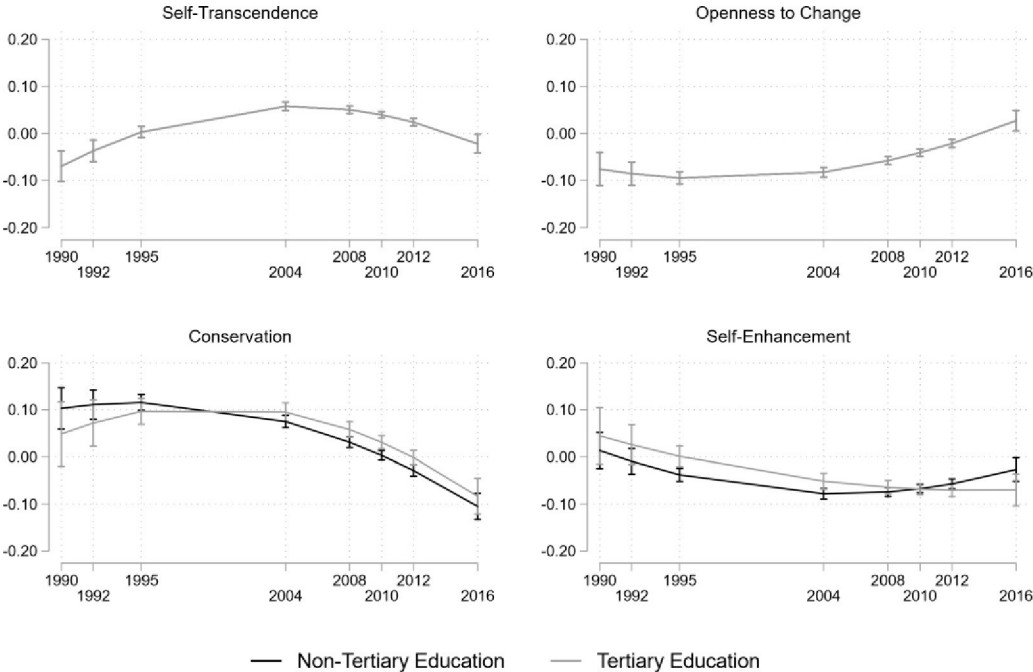

**Fig 5. Predicted slope in 4-year intervals on the HOVs on a 4-point scale in GSOEP data.** *Note*. Bars indicate 95% confidence intervals. Black lines stand for people without, grey lines for people with tertiary education.

The estimated means of conservation are shown in Fig 4. Independent of their education, respondents rate conservation similarly in 1990 (the difference is 0.042, $p = 0.253$), diverging slightly only in 2012 and 2016 (the difference is 0.088, $p < 0.001$). The ratings of this HOV start at 3.16 for the tertiary (3.11 for non-tertiary) educated participants in 1990 and increase to 3.56 (3.47) 26 years later. The slope of conservation does differ between education levels as shown in Fig 5. While being positive for non-tertiary educated respondents in 1990 ($ß = 0.33$, $p < 0.001$), it is not significant for tertiary educated respondents ($ß = 0.15$, $p = 0.164$). By 1995, the slope increases for both groups (non-tertiary: $ß = 0.36$, $p < 0.001$ and tertiary: $ß = 0.31$, $p < 0.001$). However, it decreased to become negative in 2016 for both education levels $ß = -0.33$, $p < 0.001$ (non-tertiary) and $ß = -0.26$, $p < 0.001$ (tertiary).

The importance of self-enhancement clearly decreases, as displayed in Fig 4. In 1990 it is highly rated for respondents both with non-tertiary (3.21) and tertiary education (3.12), and the difference between education levels is significant ($p = 0.004$). By 2016 this HOV decreases (2.86 vs. 2.89), and the difference between education levels is insignificant ($\Delta = 0.026$, $p = 0.145$). Fig 5 shows that the slope of self-enhancement is non-linear. For the group with lower education levels, it starts as non-significant in 1990 and 1992. In 1995, when respondents were in their mid-20s, the slope became negative ($ß = -0.14$, $p < 0.001$) and decreased to $ß = -0.29$, $p < 0.001$ by 2004, when respondents were in their early to mid-30s, but increases again ($ß = -0.10$, $p = 0.037$) by 2016. For the tertiary-educated group the slope is significantly delayed, in 2004 it lies at $ß = -0.19$ ($p < 0.001$) and increases to $ß = -0.26$ ($p < 0.001$) by 2010 when respondents are in their late 30s and stay there until 2016.

## Discussion

To summarize the results of Study 3 we indeed find evidence for a lifelong development of values which–again–contrasts present research in the field. However, it supports current literature proposing an increase in social-focused and a decrease in personal focused values with advancing age. Nevertheless, we also found that both change and stability, seem to be associated with clear stages in life. All four HOVs appear as rather stable in the earlier period of adulthood, starting around the age of 21 (measured from 1990 to 1995). Here, conservation and self-enhancement values do not differ between education levels, whereas openness to change and self-transcendence are rated as being more important by highly educated individuals over the entire lifespan. Change in all HOVs occurs particularly from the middle 20s to the middle 30s (between 1995–2008), resulting in a stabilization of importance ratings in people's late 40s.

Another interesting finding is that self-enhancement and self-transcendence values exhibit development trajectories that are less heterogeneous than conservation and openness to change values and that they change more. In terms of value priorities these trajectories indicate a shift toward a social-focused value profile, with higher educated individuals being slightly more 'growth'-oriented whereas lower educated participants are slightly more oriented toward 'anxiety avoidance'.

## General discussion

The central aim of this paper was to shed light on the question if values are really stable throughout adulthood, as proposed by various value theorists [5, 6]. Generally, the investigation of values across the lifespan is rather difficult because of a lack of appropriate longitudinal data and value measures. In the present paper we offer a solution to this problem. By applying an uncommon approach, we were able to make use of two longitudinal datasets measuring HOV orientations in line with Schwartz's TBHV over a period of more than 25 years. To our

knowledge it is the first time that a research endeavor assessed value development throughout adulthood in such an encompassing way, exempting the study by Leijen et al. [9] who examined longitudinal data spanning a period of 12 years in the life of Dutch adults.

Our studies reveal that values of adults are not stable across the lifespan—in contrast to the referenced theoretical assumptions. On the contrary, they develop continuously, only at a slower pace than earlier in life. Our data show, for example, that self-transcendence values become increasingly important with age. Though we cannot exclude a genetic predisposition toward a specific value development trajectory [22], we are confident that value development unfolds throughout life and is strongly shaped by age-related life events. This means that ever changing motivational goals will eventually change people's actions and thoughts.

Overall, we found that changes in all HOVs correspond to different stages across the lifespan. With increasing age, people become more integrated in, and to some extent dependent on, social networks and social interaction. It is therefore no surprise that self-transcendence and conservation, the two social focused values, gain importance across the lifespan, whereas the rather self-focused values of openness to change and self-enhancement lose relevance over time.

Study 3 reveals that education does play an important role in the trajectories of self-enhancement. In the mid-30s, individuals without higher education rate self-enhancement lower than those with tertiary education, reversing the relationship observed between education and self-enhancement in the early 20s. Although we can only speculate about the reasons for this development, one possibility could be that central life events occur at different stages in the lives of higher and lower educated individuals (cf. [15])—while some invest their 20's into developing a sense of self-security and personal achievement, others will attempt doing so at a later stage. Other socio-economic structural factors could also play a role: As higher education takes longer, it delays life-course transitions, such as entering the labor market, establishing a stable relationship, and having children [60, 61].

This finding strongly resonates with the central arguments of the Socioemotional Selectivity Theory (for a review see [62]), which describes a shift away from personal-oriented goals with increasing age. While younger people are regarded as being focused on personal-oriented goals, such as their careers, older people are assumed to focus more on social, high-quality contacts. The results of our studies can therefore strengthen the bridge linking human values and aging research (e.g., [63]).

Turning to conservation values, generally, the developmental trajectory in Study 3 parallels Study 2. However, education levels begin to impact the rate of change and importance of conservation in the late 30s, similar to self-enhancement. Interestingly and in contrast to current literature based on cross-sectional findings (e.g., [13]), those with non-tertiary education appear to devalue conservation with age slightly more than those with tertiary education. Put differently, we find that in the longitudinal GSOEP data, with increasing age highly educated individuals seem to value conservation more than people with lower education. One interpretation of this finding is that individuals who invest economically in a social system and derive social status from it will also defend and conform to it [64]. Future research could investigate this phenomenon further to understand how life history and pacing of events, such as education trajectories, and aging impact the importance of conservation values. Across two different samples from Germany, we showed that value developmental trajectories in adulthood are not always linear therefore providing evidence that cross-sectional (e.g., [31]) and short-term longitudinal studies [8] may not be able to accurately distinguish between age, cohort and period effects. We observed a decline in the importance of openness-to-change in both the highly educated peace activist sample (Study 2) and the general population sample (Study 3) until approximately the age of 30. This occurred in the context of the socio-economic crisis of

2007–2008, during which younger generations faced high levels of unemployment and economic instability, which may have hindered their ability to pursue a lifestyle that involves stimulating activities and independent thinking. However, whereas the peace activist sample eventually returned to positive rates of change emerging out of the crisis, the general population sample plateaued thereafter. The plateau in the general population coincides with patterns found in the general population in New Zealand [8] and the Netherlands [9]. There appears to be something unique to the peace activists in Germany however that made them experience differently the years after the crisis, and education is most likely not the answer. Perhaps their worries, or better yet the ability to find meaning in life despite having to cope with worries about an uncertain future might provide a better explanation (e.g., [65]).

Across the two samples we observed value development in line with the relations of incompatibility-compatibility of value motivational goals that are postulated in the TBHV (see Fig 1; also see [15]): Both, conservation and openness to change values, as well as self-transcendence and self-enhancement each develop in opposite directions. This finding corroborates our current understanding that values with incompatible goals do not become intertwined with age. To understand whether this finding is unique to the German population or generalizable further, replication studies with different samples are necessary, especially when considering findings by Witte at al. [66] who showed that across European countries values of more than 30% of people follow an erratic organizational structure that does not match the TBHV theory.

## Limitations and future research directions

There is one important difference between the LuNT Study and the GSOEP pertaining to the form of change over time. The GSOEP data exhibits curvilinear changes meaning that the slopes of all values except of self-transcendence were quadratic. In contrast, in the LuNT data linear changes prevail. There are several possible explanations, including the differences in scales and the sampling methods. The GSOEP data include a less accurate response scale (1 to 4) compared to the LuNT data (-1 to 7), which makes it more likely that there are larger changes in ratings. Despite the differences in form, the direction of change is the same across samples and the magnitude of change across the observation period is similar, too: Conservation changes most, followed by openness to change and self-enhancement, while self-transcendence appears to be most stable across the life span.

The second important difference between the two datasets concerns the sampling method. The GSOEP data is a representative sample of the German population and includes rigorous sampling procedures and refreshment samples that allow researchers to estimate coefficients that are generalizable to the population. In contrast, the LuNT study includes highly engaged individuals who have a strong connection to the peace movement and the LuNT panel itself. Their continued, non-renumerated and voluntary involvement in the study for over 30 years suggests that they are a particular subsample of the German population. Differences in samples are highly informative as they clearly indicate that the average trajectories estimated with the GSOEP hide substantial heterogeneity of the lifespan development of values in the population, one of which is exemplified with the LuNT panel.

Future research should investigate the heterogeneity of value development further by considering how social, demographic and economic processes unfolding over the lifespan as well as specific (historical) events impact the level and changes across the lifespan. Research has already contributed to this field; however, much is still unknown. Kohn, for example, found large differences in openness to change and conservation across social classes [41]. Value transmission studies showed differences in levels and changes of value preferences between natives and migrants [67, 68] and another strand of research found that individuals differ in

their values depending on their occupational work logic, whether they work with people, with things, or as managers and administrators ([69, 70]. However, the most likely differences in HOV levels and trajectories should be found between genders. Men and women have widely different life-courses and options, even in the most egalitarian societies [71]. Furthermore, gender can affect values from an early age onwards through socialization as well as through genetics and the interaction of both factors [22]. However, it was beyond the scope of this study to investigate socio-economic factors of value change.

Lastly, future studies can build on the present research to investigate sudden disruptions in the societal structure—such as due to the COVID-19 pandemic or economic crises—and to what extent they can cause anomalies in the value developmental curves of people (e.g., [72, 73]. Such studies would also advance our understanding of whether such anomalies have a long-lasting effect on value development curves or not. For example, the impact of some events like the financial crises or the war in Iraq may have affected the LuNT participants differently than the general population, causing the discrepancies in value development observed in openness to change values.

## Conclusion

The presented research shows that individual values never stop developing, but that compared to the early stages of life, the pace of value development slows down substantially. Our two longitudinal studies capture intra-individual value change over a much longer period (20 + years) than previous studies (2–12 years). Such data are necessary to understand intraindividual development, since cross-sectional and shorter longitudinal designs are incapable of distinguishing cohort, aging, and life stage effects. We show that conservation values become increasingly important with age, while openness to change values lose relevance. Moreover, we found that openness to change values exhibit a non-linear development, underscoring that cross-sectional research examining development on this value is rather unsuitable. At the same time, we show that with getting older the importance of self-transcendence values increases, while the relevance of self-enhancement values decreases. These results largely corroborate the hypothesis that individuals become more social-focused in values the older they get. Additionally, we find that, on average, higher educated individuals are slightly more growth-oriented than lower educated people. Our study emphasizes the importance of a true lifespan approach for the investigation of value development throughout adulthood.

## Supporting information

**S1 File.**
(DOCX)

## Author Contributions

**Conceptualization:** Adrian Stanciu, Regina Arant, Klaus Boehnke.

**Data curation:** Regina Arant, Klaus Boehnke.

**Formal analysis:** Oscar Smallenbroek.

**Methodology:** Oscar Smallenbroek.

**Visualization:** Oscar Smallenbroek.

**Writing – original draft:** Oscar Smallenbroek, Adrian Stanciu, Klaus Boehnke.

**Writing – review & editing:** Oscar Smallenbroek, Adrian Stanciu, Regina Arant, Klaus Boehnke.

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
