## [Decision Letter · Decision Letter 0]

11 Apr 2023

PONE-D-23-01126Are Values Stable Throughout Adulthood?

Evidence from two German Longterm Panel StudiesPLOS ONE

Dear Dr. Boehnke,

Thank you for submitting your manuscript to PLOS ONE. After careful consideration, we feel that it has merit but does not fully meet PLOS ONE’s publication criteria as it currently stands. Therefore, we invite you to submit a revised version of the manuscript that addresses the points raised during the review process. The reviewers particularly raise comments around the data treatment and positioning of the paper in the recent literature. 

We look forward to receiving your revised manuscript.

Kind regards,

Simon Porcher

Academic Editor

PLOS ONE

Journal Requirements:

https://journals.plos.org/plosone/s/file?id=ba62/PLOSOne_formatting_sample_title_authors_affiliations.pdf"

“Since its inception in 1985 the study referenced below as LuNT Study received occasional minor funding (one-time payments never exceeded 5000 Deutschmark) from Freudenberg-Stiftung, from the German branch of the International Physicians for the Prevention of Nuclear War (IPPNW, Nobel Laureate in 1985), as well as from the Gruner & Jahr and Der Spiegel publishing houses.”

Reviewers' comments:

Reviewer's Responses to Questions

**Comments to the Author**

1. Is the manuscript technically sound, and do the data support the conclusions?

Reviewer #1: Yes

Reviewer #2: Partly

2. Has the statistical analysis been performed appropriately and rigorously? 

Reviewer #1: Yes

Reviewer #2: No

3. Have the authors made all data underlying the findings in their manuscript fully available?

Reviewer #1: Yes

Reviewer #2: No

4. Is the manuscript presented in an intelligible fashion and written in standard English?

Reviewer #1: Yes

Reviewer #2: No

5. Review Comments to the Author

Reviewer #1: Review PLOS ONE – PONE-D-23-01126

“Are Values Stable Throughout Adulthood? Evidence from two German Longterm Panel Studies”

Oscar Smallenbroek, Adrian Stanciu, Regina Arant & Klaus Boehnke

The article is well written and deals with a current topic in the scientific literature. Unlike most previous research on value change, this manuscript uses longitudinal studies covering 18 and 26 years, respectively. Previous research is mainly cross-sectional or covers short time periods, such as 3 years. In the study, the focus is on people between 14 and 50 years of age, with a focus on the group from 28 to 40 years of age. It is an elaborate study showing with several methodologies how value change occurs in adults using three different samples.

I like the manuscript, but some revisions need to be made. A recent article on value change in Scientific Reports is missed and including the findings of this article in the current manuscript is needed. Most other remarks refer to clarification and elaboration.

This research contributes to the literature, but the claim that this is the first study “to systematically investigate whether values remain stable across the lifespan or change throughout adulthood” (p. 4) and on p. 10 (first sentence under The Present Research”) is not warranted. The article by Leijen, Van Herk and Bardi (2022) in Scientific Reports covers the same topic using a large representative sample from the Netherlands, covering 12 years (2008-2020) and includes the same individuals who were between 16 and 84 (in 2008) and were present in all 7 time-points. This article should be incorporated into the introduction and its results should be reflected upon in the results and discussion sections of the current manuscript. The Leijen et al. (2022) article has similarities to the current manuscript, and it is insightful and important to reflect on these outcomes from Germany's neighboring country in the current manuscript. It should be noted, however, that the approach taken by the current manuscript with a focus on years is different from Leijen et al. (2022) and provides novel insight.

On page 6, some references to current work on value change in children could be cited (e.g., Daniel et al., 2020).

On p. 7 the article by Leijen et al. (2022) could be included to emphasize the point that longitudinal studies indicate values change in individuals aged 16-84.

On page 11, the part on GSOEP needs elaboration. It is not clear from the text how old the individuals were in the 26 years mentioned. Please clarify.

At the end of page 12 the sample description of Study 1 is unclear. It is secondary data, so it seems to me that the authors did not make decisions on sampling. Who did and where is it reported. Please elaborate this part of the text.

At the end of page 14, it is stated that missing on more than 75% of the items was seen as problematic. Does this mean that 74% or less missing is considered fine? When that was the case, what was done? Imputing? How? What is the distribution of missing across individuals? Please clarify.

Page 17. The citation to the smacof package paper is outdated. The recent citation is Mair, Groenen, De Leeuw, 2022. This refers to the current updated package in R. See below.

p. 21 Is the “Longitudinal Internet Studies for the Social Sciences” the LISS panel?

p. 30. The last sentence is unclear: “there is no sample is assessed in each measurement wave”. Please change.

On page 31 it is written that the GSOEP is a chain of shorter panels. This is fine but it also indicates a difference with the Leijen et al. (2022) study in which the same people were included in all waves over the 12-year period. That the results based on GSOEP are in line with the findings in the Netherlands again provides evidence for the way in which values change in adulthood. Also, shorter panels provide similar results.

On page 37 at the end, I think a possible explanation for the return to a positive rate of change may be in the return of economic growth after the recession caused by the global financial crises (GFC) in 2008. People in their early 30s were then more affected by the GFC than older generations (e.g., through temporary unemployment or difficulty in getting a job) but were able to pick up quickly after the GFC.

Page 38. Change in HOV stabilizes in the late 40s. This may be related to Leijen et al. (2022) who find that values of Baby Boomers and the Silent generation hardly change over a 12-years period. Please reflect on this.

References

Daniel, E., Benish-Weisman, M., Sneddon, J.N. and Lee, J.A. (2020), Value Profiles During Middle Childhood: Developmental Processes and Social Behavior. Child Dev, 91: 1615-1630. https://doi.org/10.1111/cdev.13362

Leijen, I., van Herk, H. & Bardi, A. Individual and generational value change in an adult population, a 12-year longitudinal panel study. Sci Rep 12, 17844 (2022). https://doi.org/10.1038/s41598-022-22862-1

Mair, P., Groenen, P. J., & de Leeuw, J. (2022). More on multidimensional scaling and unfolding in R: smacof version 2. Journal of Statistical Software, 102, 1-47.

Reviewer #2: This is an amazing set of datasets and great set of analyses.

However, it is not that convincing in a number of points.

Conceptually:

The paper is currently quite repetitive and long-winded. The measures and their histories are described repeatedly, the lack of longer term longitudinal studies is stressed at length at various points, etc. At the same time, alternative models such as biological maturation, life history/life pace effects or normative pressures as well as contextual (in particular economic) changes are not mentioned (see for example Fischer, 2017 for a broad picture overview). The two longitudinal studies also overlap substantially and constant backreferencing is occurring in study 3. Possibly, the two longitudinal studies could be integrated into a single study?

It may make sense to discuss the value model in terms of person vs social focused and anxiety vs growth values. This helps to map better onto sociological theories of values and it also may make it easier to explain some patterns.

Data analysis points:

For the validation study:

a) Please use raw data for all analyses and measures (at least check whether this makes a difference)

b) Probably a better approach would be to randomly split the sample, run a mds with all items together (the pvq and svs-10 items could be set to the theoretically expected position as per bilsky et al., the K&S items could be left free to vary), then use the second set to confirm the structure. The three analyses are all conducted on the same dataset and the final analysis is not independent from the first explorations.

c) I am not sure whether this analysis shows measurement equivalence, it seems to show convergent validity for these measures in my books.

For both longitudinal studies:

How much variance in the variability is explained by these temporal effects? You could break down explained variance using the r2mlm package.

Study 3 – why not use the full data set? This would be much stronger than trying to match to the not representative sample of peace activists. I strongly recommend repeating the analyses with the full dataset.

How did you deal with possible changes in education during this period (e.g., evening school, late entry to university)?

Data interpretation question:

To what extent could these effects be cohort or event effects? For example for the peace activist cohort: Could it be that 2009 was particularly depressing economically (remember that it was around the world economic crisis and I think Germany suffered significantly around that time) and therefore resulted in lower openness, which then rebounded 10 years later? This also seems to fit the increasing conservatism (security?) concern over time. Or are there some other external factors that could influence these value scores? The Gaza war? The continuing casualties in Iraq? Somewhat similar patterns seem to appear in the representative sample. I am just trying to highlight that the analysis is not free from external forces that may influence value shifts.

For study 3: The effect of education on valuing conservation values more in mid-life makes sense evolutionary speaking. For example, at this moment in life more highly educated individuals are entering positions of power within a social hierarchy, so they have to benefit more from upholding and preserving the system (I think some of the work within social dominance theory and biological theories of social status could be useful).

Minor points:

The Milfont et al study is problematic in that they seem to find negative slopes for all values, suggesting that all values decreased over their 3 year study period. I found this hard to interpret (beyond some measurement bias effect) and if I am not mistaken, the authors did not attempt to provide an explanation either (I could misremember).

Smacof not smacoff

6. PLOS authors have the option to publish the peer review history of their article (what does this mean?). If published, this will include your full peer review and any attached files.

Reviewer #1: **Yes: **Hester van Herk, PhD

Reviewer #2: **Yes: **Ronald Fischer

---

## [Author Response · Author response to Decision Letter 0]

13 Jun 2023

An extensive rebuttal letter is included in a separate file

---

## [Decision Letter · Decision Letter 1]

20 Jul 2023

Are Values Stable Throughout Adulthood?

Evidence from two German Longterm Panel Studies

PONE-D-23-01126R1

Dear Dr. Boehnke,

We’re pleased to inform you that your manuscript has been judged scientifically suitable for publication and will be formally accepted for publication once it meets all outstanding technical requirements.

Kind regards,

Simon Porcher

Academic Editor

PLOS ONE

Additional Editor Comments (optional):

Reviewers' comments:

Reviewer's Responses to Questions

**Comments to the Author**

1. If the authors have adequately addressed your comments raised in a previous round of review and you feel that this manuscript is now acceptable for publication, you may indicate that here to bypass the “Comments to the Author” section, enter your conflict of interest statement in the “Confidential to Editor” section, and submit your "Accept" recommendation.

Reviewer #1: All comments have been addressed

Reviewer #2: All comments have been addressed

2. Is the manuscript technically sound, and do the data support the conclusions?

Reviewer #1: Yes

Reviewer #2: Yes

3. Has the statistical analysis been performed appropriately and rigorously? 

Reviewer #1: Yes

Reviewer #2: Yes

4. Have the authors made all data underlying the findings in their manuscript fully available?

Reviewer #1: Yes

Reviewer #2: No

5. Is the manuscript presented in an intelligible fashion and written in standard English?

Reviewer #1: Yes

Reviewer #2: Yes

6. Review Comments to the Author

Reviewer #1: In the revised manuscript all points I made have been satisfactorily addressed. I like the current manuscript as it has a timely topic and is well written.

Reviewer #2: Thank you for your revisions. The study reports interesting findings that raise interesting questions and guide further studies in this field.

7. PLOS authors have the option to publish the peer review history of their article (what does this mean?). If published, this will include your full peer review and any attached files.

Reviewer #1: **Yes: **Hester van Herk

Reviewer #2: No

---

## [Editor Report · Acceptance letter]

23 Aug 2023

PONE-D-23-01126R1 

 Are Values Stable Throughout Adulthood? Evidence from two German Long-term Panel Studies 

Dear Dr. Boehnke:

I'm pleased to inform you that your manuscript has been deemed suitable for publication in PLOS ONE. Congratulations! Your manuscript is now with our production department. 

Kind regards, 

on behalf of

Pr. Simon Porcher 

Academic Editor

PLOS ONE